# SHOW OR TELL? EFFECTIVELY PROMPTING VISION-LANGUAGE MODELS FOR SEMANTIC SEGMENTATION

## ABSTRACT

Large Vision-Language Models (VLMs) are increasingly being regarded as foundation models that can be instructed to solve diverse tasks by prompting, without task-specific training. We examine the seemingly obvious question: *how to effectively prompt VLMs for semantic segmentation*. To that end, we systematically evaluate the segmentation performance of several recent models guided by either text or visual prompts on the diverse MESS dataset collection. We introduce a scalable prompting scheme, *few-shot prompted semantic segmentation*, inspired by open-vocabulary segmentation and few-shot learning. It turns out that even the most advanced VLMs lag far behind specialist models trained for a specific segmentation task, by about 30% on average on the Intersection-over-Union metric. Moreover, we find that text prompts and visual prompts are complementary: each one of the two modes fails on many examples that the other one can solve. Our analysis suggests that being able to anticipate the most effective prompt modality can lead to a 11% improvement in performance. Motivated by our findings, we propose PromptMatcher, a remarkably simple baseline that combines both text and visual prompts, achieving state-of-the-art results for training-free semantic segmentation.

## 1 INTRODUCTION

Large Vision-Language Models (VLMs) have established themselves as the state-of-the-art for cross-modal reasoning that involves images and text, and even as robust backbones for purely visual tasks, benefiting from the wealth of semantic and contextual relations contributed by language modeling. A particular strength of VLMs is the capability to condition image understanding on text inputs, the so-called *Text Prompts* (TP). This enables, for instance, segmentation of a specific object in an image (Lai et al., 2024; Rasheed et al., 2024), reasoning about relations between objects (You et al., 2023; Peng et al., 2023), and visual question answering (Beyer et al., 2024; Xiao et al., 2023). Some VLMs also offer conditioning on *Visual Prompts* (VP). Typically these are visual cues like points (suitably embedded coordinates on the image), scribbles or bounding boxes (Lai et al., 2024; Rasheed et al., 2024), but it has also been proposed to directly superimpose symbols in pixel space (Yang et al., 2023a).

We observe that (prompted) VLMs have been studied mainly in two broad settings. The first one could be called *image-driven text generation*, meaning that the system outputs language, while visual information is used only on the input side. This setting includes tasks such as image captioning and visual question answering. The second setting can be referred to as *visual grounding*. This setting links language to image regions, helping to enhance the model's spatial reasoning and understanding of how textual descriptions correspond to visual elements in an image. Examples include phrase grounding, where the model is asked to detect the objects mentioned in the text, constraining their spatial relations, and referring expression comprehension, where objects have to be identified based on a periphrasis, thus emphasising contextual relations.

In this work, we focus on the potential of prompting mechanisms to improve image-to-image tasks. Given that large VLMs are increasingly being recognized as foundation models for vision, we ask how to effectively prompt VLMs for semantic segmentation. In other words, our primary interest is not how well the model can parse or generate text about images, but rather how accurately it can delineate objects in images.

Since the desired outputs – segmentation masks – reside in image space, it is a natural question whether Text Prompts or Visual Prompts are more expedient, and how the two can be combined. While text prompting has proved successful in guiding image understanding and visual reasoning, we claim that *it is not always sufficient to prompt a VLM with text*, and *visual prompts can in some cases be more suitable, or complementary*. Intuitively, a visual example can in certain situations convey information that it much harder, or even impossible, to transmit through text. While the internal mechanisms of large models are notoriously difficult to disentangle and interpret, there is a simple argument in support of visual prompting: The *projection* of the visual world to language is lossy. Even elaborate text descriptions are often ambiguous and can lead to vastly different predictions.

At this point we must highlight a subtle, but important difference that is sometimes overlooked: text prompts are normally understood as generic statements that can be defined once and then applied across many images, like "segment all cats". In contrast, visual prompts are predominantly understood as image-specific, like for instance a scribble to denote the cat in a particular image. In this interpretation, visual prompting requires user input for every new sample and is not scalable. Instead, we advocate for a form of visual prompting that incurs only a constant overhead for arbitrarily large test sets: The user annotates instances of their desired target class on a small number of images, then that fixed set of examples serves as the prompt for the full dataset and no further interaction is expected. We refer to this setup as *few-shot prompted semantic segmentation* (FPSS). Unlike traditional few-shot learning, which also uses a small set of annotated examples but requires fine-tuning the model, FPSS operates through prompting rather than training. It is also related to *open-vocabulary segmentation*, where a frozen model is adapted to new classes without retraining, though typically in a zero-shot context rather than using a few-shot approach.

When evaluating under the FPSS protocol, we find that VLMs are not behaving (yet) as *foundational*. They still trail domain-specific segmentation models by about 30% on average in Intersection-over-Union (IoU) score on the dataset used in this work. Furthermore, we find that text prompts perform better *on average*, but that visual prompts are able to address tasks that are exceptionally difficult for text prompted models. Unsurprisingly, the two prompting modes are to some degree complementary: in hard scenarios, e.g. medical imaging, VP can solve many instances that TP cannot, and vice versa.

Motivated by these findings, we construct a simple baseline for combined text and visual guidance, while still maintaining a training-free, prompting-only setup. Prompting with both text and vision indeed improves the performance by a significant 2.5% compared to only text (respectively, 3.5% compared to only vision).

Summarizing our contributions:

- We design a novel benchmarking task to probe the performance of VLMs as semantic segmentation engines.
- We show that even the latest models remain far below custom models trained for a specific task and data domain. In other words, we are still far from *foundational* VLMs.
- We show that text and visual prompting complement each other, and that being able to anticipate the most effective prompt modality can lead to a 11% improvement in performance.
- We propose a simple training-free framework to capitalize on the complementary strengths of text and visual prompts and achieve state-of-the-art on the MESS dataset collection Blumenstiel et al. (2023).

## 2 TASK FORMULATION

The goal of our paper is to evaluate to which extent (training-free) prompting of generalist VLMs can replace specialist models for semantic segmentation. It is obvious that some form of prompt is always required to let a VLM know what to segment, but it is much less obvious what the most suitable prompt is. Here, we limit ourselves to the two most popular ones, text and visual prompts.

As an example, let us assume we want to segment airplanes. A natural way to instruct the model is with one or a few text prompts, like "segment all airplanes". Note that, due to the compositional nature of language, there is no clear definition on how many prompts we are effectively using, since two or more prompts can be merged into one, as in "segment airplanes and similar flying machines".

In normal text prompting, the same prompt is then applied to all input images. FPSS translates that one-off prompting scenario to the visual domain: the user supplies the system with at most $K$ reference images of airplanes, along with their segmentation masks or other annotations (e.g., a set of points within the mask). Based on that input, the system shall segment airplanes in any number of unseen target images. Note that this mode of interaction makes it possible to communicate about visual concepts whose category name is not known to the model, just like a child can say "I want this" before learning the word "chocolate".

Beyond the research questions on how the two prompting modes compare and when one or the other is more successful, prompting in the FPSS setting is relevant in several real application scenarios as digitalization and AI permeate society. For instance, an engineer may have to instruct an inspection system to examine a new item, or a biologist may want to screen a legacy image collection for a newly discovered species; In both scenarios, users may prefer to provide only a few text or visual prompts to the system, expecting the task to be automatically applied to the entire dataset.

## 3 ANALYSIS

In this section, we outline the evaluation framework, specifying the models considered within FPSS, specifically under the one-shot regime. In particular, we select a range of key text prompted and visual prompted models and assess their effectiveness in performing segmentation when provided with the corresponding prompt modality. We then present and discuss the results, providing a detailed analysis of the performance differences across modalities, highlighting strengths and limitations.

### 3.1 EVALUATION PROTOCOL

There are many models capable of performing segmentation guided by text prompts, mainly falling into two categories: open-vocabulary segmentation models (Cho et al., 2024) and vision-language models (VLMs) (Lai et al., 2024; Beyer et al., 2024). Both types of models leverage textual input to guide segmentation, with open-vocabulary models focusing specifically on identifying objects beyond a fixed set of categories, while VLMs, with their broader multi-modal capabilities, can also be adapted for segmentation tasks. Similarly, we identify two categories of models that can be prompted visually: models specifically trained with visual prompts (Li et al., 2023a; Zou et al., 2023) and training-free frameworks leveraging existing segmentation models along with matching algorithms (Liu et al., 2024b; Frick et al., 2024). In contrast, very few models have been presented that can be guided with both text and visual prompts (Zou et al., 2023)

For open-vocabulary segmentation models, we consider CAT-Seg (Cho et al., 2024), the state-of-the-art on the MESS dataset. In particular, we use CAT-Seg with the *CLIP ViT-L/14* backbone. We also include SEEM (Zou et al., 2023), specifically the SEEM *Davit-Large* implementation. This is the only available model to accept TPs and VPs simultaneously, although in this section we only use them separately. Combined prompting with SEEM is discussed in Section 5.

As VLM baselines, we include the decoder-free Florence-2 (Xiao et al., 2023), specifically the segmentation branch of the large, fine-tuned model, where we clip the generated sequence length to 1024 for computational reasons; and PALI-Gemma (Beyer et al., 2024), a small but effective architecture using a VQVAE decoder van den Oord et al. (2018). Regarding PALI-Gemma, we make use of the standard *224-mix* implementation. We also evaluate the recent LISA (Lai et al., 2024), in particular the *LISA-13B-llama2-v1* version, which features a dedicated decoder (from the SAM foundation model). To keep the evaluation focused, and taking computational resource limitations into account, we regard LISA as proxy for its follow-up works: GLAMM (Rasheed et al., 2024) and SESAME (Wu et al., 2023), which might offer marginal improvements. Our choice of VLMs is primarily informed by their referring segmentation performance on the RefCOCO, RefCOCO+, and RefCOCOg datasets (Kazemzadeh et al., 2014; Mao et al., 2016), a task which is closely related to our FPSS task. In all cases, we opt for greedy LLM decoding.

When considering models which are specifically trained with visual prompts, we once more pick SEEM (Zou et al., 2023), using the same implementation as described for the text prompting setting, as well as DINOv (Li et al., 2023a), using its Swin-L variant. Regarding visually prompted training-free frameworks, we choose Matcher (Liu et al., 2024b) motivated by its performance on COCO-20i, and its follow-up work SoftMatcher (Frick et al., 2024) mainly for its computational efficiency, both

of which leverage pre-trained foundation models, namely Segment Anything (SAM, Kirillov et al., 2023) and DINOv2 Oquab et al. (2024), in combination with traditional matching algorithms to provide image-prompted segmentation capabilities. Furthermore, we modify the Matcher/SoftMatcher framework to obtain an improved version, which we call SoftMatcher+. It utilizes AM-RADIO (Ranzinger et al., 2024) as its backbone instead of DINOv2, leveraging the excellent abilities of AM-RADIO features (distilled from several large models including CLIP, DINOv2 and SAM) in terms of matching, pixel-level localization, and vision-language connections. For all these training-free methods we make use of the ViT-L versions of the models (DINOv2, SAM, AM-RADIO), and tune their hyper-parameters on COCO-20i.

Regarding text prompts, we proceed as follows: for open-vocabulary segmentation models that accept only a class name as input, we use class names based on the dataset specifications. For VLMs with advanced language abilities, we embed the class name in the sentence "Segment all the instances of class `class_name` in the image". As visual prompts, we sample one single image of the target class from the dataset itself, together with its ground truth segmentation mask. Considering a prompt consisting of a single image is proportionate with our elementary text prompts. Picking that image from the same dataset corresponds to the realistic scenario where the user creates the prompt on images acquired in their application setting, with similar imaging conditions and class definitions as the test data. To minimise biases due to the choice of prompt image, we sample a different prompt image for each prediction.

We point out that both text prompts and visual prompts can be refined by prompt engineering. This field explores various techniques, ranging from single prompt optimization (Zhou et al., 2023), prompt ensembling (Wang et al., 2023c), to multi-step reasoning (Wei et al., 2023; Yao et al., 2023; Zhang et al., 2024b). While prompt engineering can make a substantial difference, it has become an art in itself, and in fact an entry barrier for inexperienced users. It goes beyond the scope of the present work, but may be an interesting avenue for future research.

We also consciously refrain from any fine-tuning. Often, even large models are fine-tuned for specific tasks, which can significantly improve their performance. However, in our view, this approach seems misaligned with the definition and purpose of a "foundation model", which should ideally be usable with minimal intervention. Once the hardware, data, and expertise for fine-tuning are required, there is arguably little qualitative difference from the well-established practice of training a dedicated model starting from pre-trained weights (e.g., from ImageNet).

As a testbed for our experiments we use the MESS dataset collection (Blumenstiel et al., 2023). It consists of 22 different segmentation datasets that span a wide variety of application domains and image characteristics. The datasets are grouped into five broad domains, *General* (6 datasets), *Earth* (5), *Medical* (4), *Engineering* (4) and *Agriculture* (3) as detailed in Table 5. The MESS collection is deliberately designed as a challenging benchmark for foundation models and open-vocabulary models, because its constituent datasets span a wide range of target categories and image characteristics, many of which differ significantly from the dominant conditions of scraped internet data used to train most VLMs. Moreover, MESS comes with strong baselines generated with per-dataset, domain-specific semantic segmentation models. For clarity of presentation, we always show average numbers for the five broad domains covered by MESS. The detailed dataset composition is provided in Appendix A.

The evaluations were run on a single A100 with 40GB of memory, which takes ≈14 hours for one complete run with the largest model (LISA-13B). Open-vocabulary segmentation models are faster, completing one evaluation cycle in 9 hours, while Florence-2 is the slowest, taking almost 24 hours. Visually prompted models are substantially lighter (up to 1.2B parameters) than their text prompted counterparts (up to 13B parameters), and while Matcher is very slow (22 hours), SoftMatcher+ takes around 5 hours for an evaluation cycle.

## 3.2 RESULTS

Table 1 showcases the results under the FPSS evaluation scenario on the MESS dataset. Notably, we see that all the evaluated promptable models still trail domain-specific segmentation models by about 30% IoU on average.

|  | General | Earth | Medical | Engineering | Agriculture | Average |
|---|---|---|---|---|---|---|
| SEEM text | 35.9 | 36.8 | 28.9 | 13.9 | 44.5 | 32.0 |
| CAT-Seg | 33.9 | 36.9 | **45.7** | **48.4** | 24.5 | 37.9 |
| Florence | 14.0 | 13.9 | 13.1 | 7.3 | 7.6 | 11.2 |
| PALI-Gemma | 35.3 | 29.1 | 28.4 | 7.2 | 40.0 | 28.0 |
| LISA | **57.0** | **47.6** | 31.6 | 12.7 | **63.9** | **42.6** |
| SEEM Vision | 9.6 | 16.8 | 20.5 | 6.9 | 21.7 | 15.1 |
| DINOv | 37.4 | 28.0 | 24.2 | 8.3 | 59.1 | 31.4 |
| Matcher | 43.2 | 31.2 | 26.0 | 12.4 | 54.9 | 33.5 |
| SoftMatcher | 48.0 | 34.0 | 31.5 | 18.8 | **59.8** | 38.4 |
| SoftMatcher+ | **54.1** | **35.2** | **33.4** | **25.6** | **59.8** | **41.6** |
| Supervised | 55.2 | 71.4 | 82.6 | 89.4 | 62.8 | 72.3 |

Table 1: Evaluation results on the MESS dataset. The table presents performance metrics for visual-prompted models (first block), text-prompted models (second block), and supervised baselines (last row).

In the second block of Table 1, we see that among text prompted models, CAT-Seg and SEEM remain competitive baselines when compared to the VLM approaches. In fact, with the exception of LISA, the LLM-based methods underperform relative to these baselines. We hypothesise that this performance is attributed to mainly two factors. First, the detokenization procedure employed by these models could lack the granularity required for dense tasks. Second, the training data for these models encompasses a broad range of image reasoning tasks beyond segmentation, including visual question answering, object detection, and visual grounding. This diversity in training, while beneficial for general-purpose applications, may dilute the models' effectiveness on segmentation tasks.

Moreover, LISA emerges as the front-runner, with an average IoU of $42.6\%$, around $4.5$ IoU points higher than the second best performing model CAT-Seg. This is likely due to LISA's specialized foundation model decoder and to its extensive training regimen on the large segmentation dataset SA-1B (Kirillov et al., 2023), which is then further aligned with segmentation-specific datasets such as RefCOCO or ADE20K (Zhou et al., 2018). More interestingly, comparing LISA with domain-specific models trained on individual datasets yields an important finding: we find that in some cases, LISA outperforms the baseline on generalist tasks, surpassing specialized segmentation models optimized for in-domain performance. However, it is also crucial to note that LISA's performance significantly decreases in more technical domains, such as engineering and medical applications. In these specialized areas, it is surpassed by the open-vocabulary segmentation models, particularly CAT-SEG, and by domain-specific models. This performance gap in technical domains suggests potential for improvement.

The second block of Table 1 presents the results of the visual prompted models. We see that these models underperform on average compared to their text prompted counterparts. For instance, the performance of SEEM Vision is significantly inferior to SEEM Text. And while SoftMatcher narrows this performance gap, SoftMatcher+ demonstrates even better results, nearly reaching LISA's performance level. In particular, we highlight that SoftMatcher+ shows superior performance compared to LISA on the technical domains. We attribute this improvement to the nature of image examples, which more precisely and effectively capture the user's interests with better precision and varying levels of detail.

## 4 SHOW OR TELL?

Our findings in Section 3.2 suggest that visual prompting and text prompting behave differently when it comes to different target domains. To gain deeper insights into this performance disparity, we conduct a more thorough examination of the top-performing models from each category. This comparative analysis helps us elucidate the factors underlying the performance differences between visual and text-based prompting.

|                 | General | Earth | Medical | Engineering | Agriculture | Average |
|-----------------|---------|-------|---------|-------------|-------------|---------|
| SoftMatcher+    | 53.0    | 36.2  | 30.4    | 28.7        | 60.7        | 41.8    |
| LISA            | 57.0    | 47.7  | 31.7    | 12.8        | 64.0        | 42.6    |
| Oracle Ensemble | 60.9    | 47.8  | 40.4    | 28.7        | 65.4        | 48.6    |
| Oracle Ensemble+| **67.3**| **51.8**| **46.2**| **32.5**   | **71.4**    | **53.8**|
| Supervised      | 55.3    | 71.4  | 82.6    | 89.5        | 62.8        | 72.3    |

Table 2: Oracle ensemble methods compared to the best performing text and visual prompt models, and to the supervised baseline.

| Class name         | IoU TP | IoU VP | IoU Difference |
|--------------------|--------|--------|----------------|
| Worm-eating Warbler| 1.4    | 82.2   | 80.8           |
| Rape               | 19.2   | 80.0   | 60.8           |
| Fjord              | 24.1   | 81.2   | 57.0           |
| Date               | 0.1    | 52.0   | 51.9           |
| Hair               | 18.8   | 62.1   | 43.2           |
| Upper clothes      | 16.0   | 58.2   | 42.2           |
| Tea                | 29.9   | 70.5   | 40.6           |
| Soy                | 37.2   | 77.2   | 40.0           |
| Cashew             | 27.7   | 66.9   | 39.1           |
| Kiwi               | 37.3   | 76.3   | 39.0           |

Table 3: Top 10 classes with the highest IoU difference between the text and visual prompt models.

### 4.1 ORACLE ENSEMBLING OF TEXT AND VISUAL PROMPTS

A natural starting point for characterizing the differences between visual and text prompting is to determine by how much the segmentation performance improves by choosing the best prompting modality *within each target domain*. Regarding the MESS datasets, this can be easily quantified by taking the maximum across VP and TP performance for each dataset, obtaining what we call an *Oracle Ensemble*. Table 2 shows that being able to choose optimally between using visual or text prompts brings a boost to the overall performance by 6% compared to LISA.

Motivated by this, we add more granularity to this analysis and investigate the performance upper bound that we could reach by selecting the best prompting on a *per-image* basis, as opposed to *per-dataset* ( Oracle Ensemble). We denote the resulting optimal selection with *Oracle Ensemble+* and note in Table 2 its remarkable performance of 53.8%, corresponding to an 11% jump over pure text prompting with LISA.

The simple baselines given by these Oracle Ensembles show the potential advantages of using visual prompts in conjunction with conventional text prompts. In addition, given their simplicity, they highlight the possibility that more advanced models, with access to both modalities, could achieve even greater performance when coupled with a smart integration of both sources. This motivates us to seek ways to leverage visual prompting in text prompted VLMs.

To optimally leverage visual prompts we first investigate the source of its relative advantage over text prompts. Looking at IoU differences on a per-class basis and ranking them based on the absolute difference as shown in Table 3, we uncover a striking trend. The top 10 values all favor VP, with some classes showing a remarkable performance advantage of up to 80%. This substantial disparity underscores the significant superiority of visual prompting over text prompting for certain classes, suggesting that visual cues provide a more effective means of guiding the model's segmentation process in these instances.

This analysis across different class names suggests that the shortcomings of text prompted models are not primarily due to an inability to segment specific objects, but rather stem from the nature of the prompts themselves. The classes where LISA performs poorly fall into two main categories: ambiguous descriptions such as *Upper clothes* and highly specific, uncommon class names such as *Worm-eating warbler* or *Fjord*. These findings suggest that the model's difficulties arise from

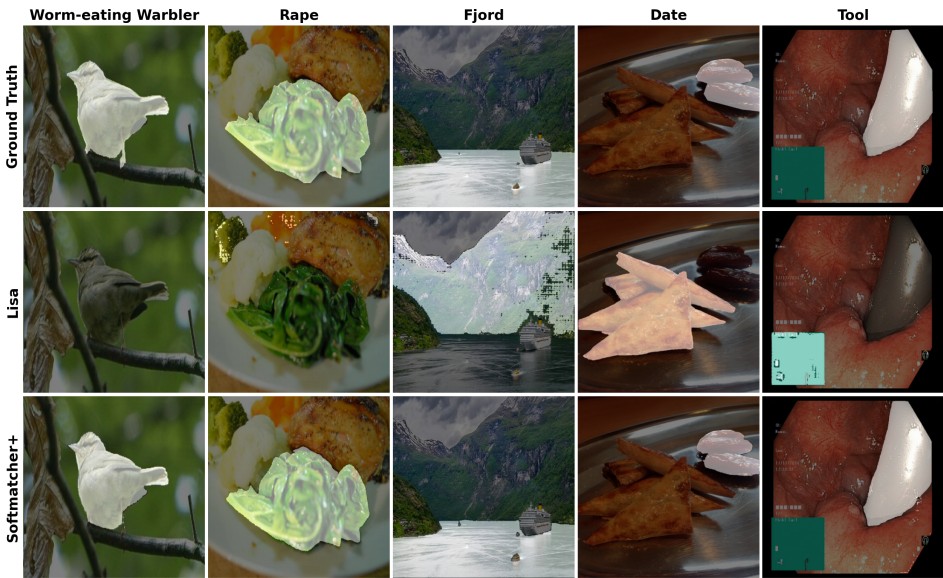

Figure 1: Qualitative analysis of the results of LISA and SoftMatcher+ compared to ground truth. The first four columns display images selected according to biggest difference of IoU between VP and TP as per Table 3. The last column displays the *Tool* class.

interpreting vague or extremely niche text prompts, rather than from fundamental limitations of its latent image encoding.

To better understand the performance discrepancies, we visually inspect samples from the first four categories, i.e. samples representing the most divergent IoU scores per class. The qualitative results can be seen in the first four columns of the Figure 1. On the first sample of class *Worm-eating warbler*, the model clearly struggles to interpret the user's request, failing to connect the specific subclass to the broader *bird* category, despite the relative segmentation-friendly image content. On the second sample, the model produces only noise at the top of the image, demonstrating a complete failure to identify the requested class of Rape (referring to the Rapeseed plant). The third sample reveals the model's confusion between segmenting the mountain portion of the fjord and the fjord itself, resulting in an inaccurate segmentation of the mountain. In the fourth example, LISA exhibits hallucination, segmenting an unrelated object when asked to segment the class *Date*.

## 4.2 AMBIGUITY OF TEXT PROMPTING

The visual inspection of the top samples in terms of performance difference between TP and VP suggests that the discrepancies can be attributed to two main linguistic challenges: ambiguity from polysemous or homonymous words and the use of highly specialized or uncommon terms.

These issues are closely related to the inherent complexities of language, which complicate the ability of text prompted systems to accurately interpret visual tasks. The interplay between ambiguity and specificity in language is inherent on how it was formed (Riemer, 1949) and it is widely known to be an issue in the computational semantics literature, hindering the algorithmic performance (Church & Patil, 1982; Manning & Schutze, 1999). The trade-off between the usage of ambiguous words and ones that are specific, unusual, or difficult to pronounce serves a crucial role in our ability to convey complex thoughts and adapt to diverse communicative contexts (Wasow, 2015).

Our hypothesis that language ambiguity can be a considerable weakness for visual prompting is supported by further experiments on the MESS FoodSeg103 dataset. Here we see a significant performance gap of 13% of IoU between Oracle Ensembling (which in this case refers to LISA) and Oracle Ensembling+. This can be attributed to the linguistic challenges previously discussed. FoodSeg103 encompasses a diverse set of food categories, many of which are either ambiguous or

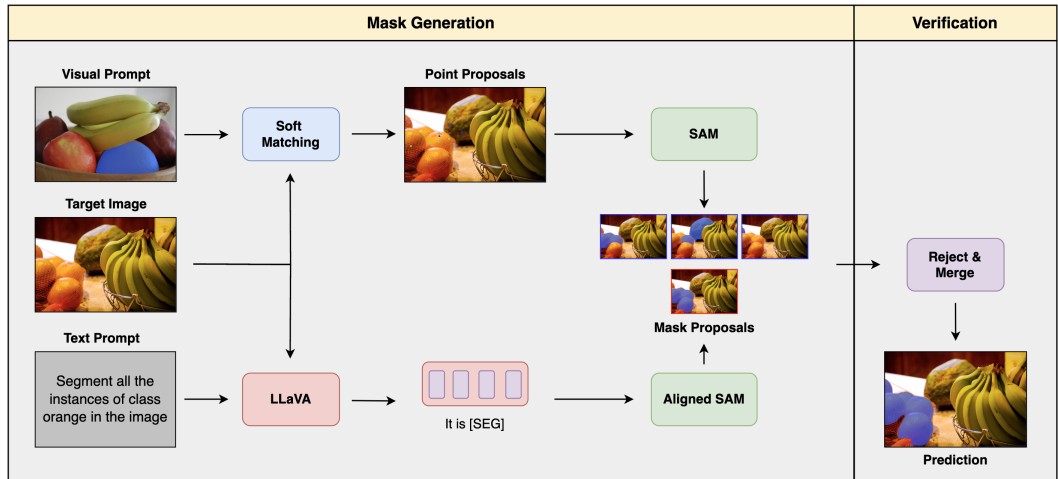

Figure 2: PromptMatcher framework: The left section illustrates the mask generation process using visual and text prompts, while the right section shows the verification module which discards inaccurate predictions.

highly specific, making them challenging to distinguish through text description. On the other hand, these foods often appear visually similar. Additional examples are provided in Appendix B.

Similarly, the Kvasir-Inst. dataset shows a notable discrepancy, particularly for the class *tool*, which is the sole category within this dataset. Examining the last column of Figure 1, we observe that the model's performance is compromised by both the non-specific nature of the word *tool* and out-of-domain nature of the image. The generality of the term *tool* sometimes leads to misinterpretation, with the model confusing it with elements of the camera interface itself. This ambiguity helps explaining the substantial 35% performance gap observed in this dataset.

Humans typically bridge this semantic gap by providing additional context (Pimentel et al., 2024). However, in our experimental setup, this approach can be prohibitively expensive or unfeasible, as shown by the *Worm-eating Warbler* case. While using the prompt "bird" could disambiguate this specific image, such generic prompts fail when working with datasets that include different bird species. Visual Prompting offers a solution to this challenge by providing a simpler, less ambiguous method to fill this semantic gap, eliminating the need for elaborate textual descriptions or context-dependent prompts.

Our considerations indicate that visual and text prompting are inherently complementary, and that visual prompting offers a natural and readily available strategy to make up for the weaknesses of text prompting due the identified ambiguities.

## 5 PROMPTMATCHER: COMBINING TEXT AND VISUAL PROMPTS

Motivated by the complementary nature of text and visual prompts, we propose a framework that effectively integrates both, closing the gap between the baselines presented in Section 3 and the Oracle Ensemble+. Furthermore, drawing inspiration from LLM-Modulo frameworks outlined in (Kambhampati et al., 2024), particularly from the concept of employing critics/verifiers to enhance generative models' reasoning capabilities, in our context we propose to use SoftMatcher+ as an effective critic/verifier for LISA's predictions. This verification module would be able to mitigate LISA's hallucinations, thereby enhancing overall accuracy.

We refer to our training-free framework as *PromptMatcher*. As depicted in Figure 2, it employs Soft-Matcher+ as both a critic and segmentation model, generating predictions using LISA for the text prompt branch and SoftMatcher+ for the visual prompt branch. First, at the mask generation step, the text prompt is processed by LISA's multi-modal LLaVA model, producing an output sequence with a specialized [SEG] token, which is then decoded into a segmentation mask by LISA's aligned SAM

| | General | Earth | Medical | Engineering | Agriculture | Average |
|---|---|---|---|---|---|---|
| SEEM | 9.7 | 17.0 | 20.5 | 7.3 | 22.5 | 15.4 |
| LISA | 57.0 | 47.7 | 31.7 | 12.8 | **64.0** | 42.6 |
| SoftMatcher+ | 53.0 | 36.2 | 30.4 | 28.7 | 60.7 | 41.8 |
| PromptMatcher | **58.7** | **39.7** | **35.1** | **30.4** | 62.4 | **45.3** |
| Oracle Ensemble+ | **67.3** | 51.8 | 46.2 | 32.5 | **71.4** | 53.8 |
| Supervised | 55.3 | **71.4** | **82.6** | **89.5** | 62.8 | **72.3** |

Table 4: Comparison of PromptMatcher's performance with i) SEEM using both visual and text prompts simultaneously ii) the top-performing text and visual prompt models, and iii) the Oracle Ensemble+ and the supervised baselines.

model. Simultaneously, SoftMatcher+'s matching pipeline processes the visual prompt, generating multiple sets of point prompts representing potential object locations. The SAM mask-decoder uses these prompts to create unique output masks for each set. Subsequently, in the verification step, we apply SoftMatcher+'s mask rejection pipeline on masks produced by both branches to verify their consistency with the reference image. This only allows plausible masks to pass, therefore playing the crucial role of a critic, reducing hallucinations originating from either branch. Finally, the verified masks are combined by taking their union to form a single, comprehensive semantic segmentation output.

We present our results in Table 4, and refer to Table 9 in the Appendix C for per-dataset results. Our combination of visual and text prompts significantly outperforms the vision-language SEEM baseline, which performs nearly the same as its vision-only version. We see that with our straightforward, training-free approach, it is possible to go beyond text-only or visual-only prompting and start to bridge the gap towards the Oracle Ensemble+. Notably, PromptMatcher surpasses Oracle Ensemble+ on two MESS datasets (DeepCrack and MHP v1), indicating synergies beyond simply selecting the better of two prompts. This superior performance can be attributed to the unique nature of the proposed framework. As our approach leverages the complementary strengths of LISA and SoftMatcher+ to generate a more diverse set of predictions, when the outputs from the two models diverge, taking their union allows merging segments from different instances. This enables the models to combine their predicted masks rather than being limited to choose the output from one or the other, which is advantageous compared to an oracle-based selection. Moreover, applying the mask rejection procedure from SoftMatcher+ to LISA masks helps to mitigate potential hallucinations from LISA by rejecting results that do not match with the reference mask. The rejection of LISA masks capitalizes on the inherent text-vision knowledge distilled into the AM-RADIO representations, improving over vision-only backbones.

Our remarkably simple integration of TPs and VPs demonstrates the immediate benefit of combining the two modalities. We are convinced that there is untapped potential in such modular, training-free frameworks. We leave the exploration of more elaborate framework designs to future work, encouraging the research community's involvement in this effort.

## 6 RELATED WORK

**Open-Vocabulary Segmentation Models** are able to perform segmentation across unlimited classes without relying on a fixed set of categories defined during training. These models often rely on CLIP-like text encoders to associate visual data with text descriptions. Specialized models like L-SEG Li et al. (2022) and CAT-Seg Cho et al. (2024) are designed specifically to solve this task, while multi-modal models such as X-Decoder Zou et al. (2022) and SEEM Zou et al. (2023) expand this capability by handling a different range of visual prompts.

**Vision-Language Models** bridge the gap between visual perception and natural language understanding, excelling in tasks that require a combination of both, such as perception-language tasks and grounding tasks. These models are built using large language models (LLMs) integrated with vision encoders. With respect to perception-language tasks, VLMs perform tasks like image captioning, visual question answering, and region-level annotations. The LLaVA series Liu et al. (2023b;a;

2024a) has set benchmarks in this area by combining vision encoders like CLIP Radford et al. (2021) with LLMs, such as LLaMA Touvron et al. (2023); et al. (2023) or Vicuna Chiang et al. (2023). InstructBLIP Dai et al. (2023) builds on the BLIP-2 Li et al. (2023b) model with instruct tuning, and MM1 McKinzie et al. (2024) provides insights into crafting effective multimodal models. GPT-4V OpenAI (2024) currently sets the highest standard in these perception-language tasks Yang et al. (2023b). In grounding tasks, VLMs are able to handle phrase grounding and referring expression comprehension, detection, and segmentation. These tasks require identifying specific objects or regions based on text descriptions. Models like Florence-2 Xiao et al. (2023) predict segmentation coordinates in the form of text, while PALI-Gemma Beyer et al. (2024) uses a next-token prediction method encoding outputs to a fixed token dictionary, which is then decoded using a VQVAE van den Oord et al. (2018). Other significant contributions include Kosmos-2 Peng et al. (2023), which integrates coordinate tokens into the vocabulary for object detection, Ferret You et al. (2023), which incorporates dense visual prompts, and Osprey Yuan et al. (2024), which adds further granularity to input prompts. While GPT-4V has shown impressive capabilities in many visual-language tasks, it has notable limitations in performing segmentation. Some VLMs incorporate specialized segmentation decoders, such as LISA Lai et al. (2024), which extends the LLaVA architecture incorporating SAM Kirillov et al. (2023) to convert predicted tokens into segmentation masks. This hybrid approach has been refined by models like GLAMM Rasheed et al. (2024), which includes pixel-level visual prompting and supports multi-round conversations, and GSVA Xia et al. (2024), which enhances resilience to adversarial attacks. PixelLM Ren et al. (2024) introduces a lightweight segmentation decoder, while SESAME Wu et al. (2023) focuses on mitigating hallucination in segmentation tasks.

**Visual Prompting** involves providing visual cues to guide the model's understanding and segmentation of images. Early works such as Bar et al. (2022), focused on solving few-shot vision tasks by reconstructing the target via image inpainting of a grid-like input prompt. This concept was further developed in models like Painter Wang et al. (2023a) and SegGPT Wang et al. (2023b), which demonstrated the possibility of solving tasks like segmentation more effectively. A significant leap forward came with the introduction of the Segment Anything Model (SAM) Kirillov et al. (2023) and its follow-up Ravi et al. (2024), showing remarkable zero-shot capabilities in image segmentation tasks. These models, along with works like OMG-LLaVA Zhang et al. (2024a), focused on using visual prompts within the target image itself, rather than relying on separate example images. Other notable works include DINOv Li et al. (2023a), which expands visual prompting from SEEM, and Matcher Liu et al. (2024b) which brings a unique approach that enables zero-shot models like SAM to be prompted one-shot through feature matching. SoftMatcher Frick et al. (2024) further expands on this concept by enhancing both simplicity and computation performance of the approach. Additionally, there has been growing research on optimizing information extraction from target images using pixel-level deformations. A seminal work in this direction is SoM Yang et al. (2023a), which posited that providing visual clues to a VLM can significantly enhance its performance. This has sparked numerous follow-up studies, including ViP-LLaVA Cai et al. (2024) that applies these concepts to models like LLaVA. The practical implications of these approaches are also being explored, such by the work He et al. (2024) in the context of web-based applications.

## 7 CONCLUSION

In this work, we introduced a benchmarking task designed to evaluate the performance of Vision-Language Models (VLMs) as semantic segmentation engines. Our results demonstrate that, despite the advancements, the latest VLMs still fall significantly short compared to custom models trained specifically on a given domain. This finding suggests that there is still room for progress in developing VLMs. We also showed that text prompting and visual prompting are complementary. By anticipating and selecting the most effective prompting modality, it is possible to achieve a notable 11% IoU performance improvement. Building on this insight, we introduced a straightforward, training-free framework that leverages the complementary strengths of both text and visual prompting, with a key verification component responsible for rejecting incorrect segmentation masks. This framework sets a new state-of-the-art benchmark on the MESS dataset collection, achieving 45.5% average IoU. Our findings highlight the potential of using multiple prompt modalities to enhance the performance of VLMs without the need for additional training, bringing us closer to true foundation VLMs.

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
