| General | ATLANTIS (Erfani et al., 2021), BDD100K (Yu et al., 2020), Dark Zurich (Sakaridis et al., 2019), DRAM (Cohen et al., 2022), FoodSeg103 (Wu et al., 2021), MHPv1 (Li et al., 2018) |
|---|---|
| Earth | FloodNet (Rahnemoonfar et al., 2020), iSAID (Zamir et al., 2019), ISPRS Potsdam (Rottensteiner et al., 2012), UAVid (Lyu et al., 2020), WorldFloods (Mateo-Garcia et al., 2021) |
| Medical | CHASE DB1 (Fraz et al., 2012), CryoNuSeg (Mahbod et al., 2021), Kvasir-Inst. (Jha et al., 2021), PAXRay-4 (Seibold et al., 2022) |
| Engineering | Corrosion CS (Bianchi & Hebdon, 2021), DeepCrack (Liu et al., 2019), PST900 (Shivakumar et al., 2019), ZeroWaste-f (Bashkirova et al., 2022) |
| Agriculture | CUB-200 (Wah et al., 2011), CWFID (Haug & Ostermann, 2015), SUIM (Islam et al., 2020) |

Table 5: Grouping of datasets in the MESS collection (Blumenstiel et al., 2023).

## A  MESS DATASET COMPOSITION

MESS Dataset integrates 22 datasets selected for their unique challenges, grouped into General, Earth, Medical, Engineering, and Agriculture domains. It evaluates model performance on out-of-distribution and adversarial examples, featuring visually complex medical images like those in Kvasir-Inst., and granular subclass divisions of common categories as seen in FoodSeg103 Wu et al. (2021) and Caltech-UCSD Birds Wah et al. (2011) datasets. Table 5 displays the dataset grouping breakdown.

## B  EXTENDED QUALITATIVE ANALYSIS

Figure 3 showcases additional examples where LISA encounters difficulties with certain classes in FoodSeg103. These images are selected from specific categories that proved challenging for the model. In the first image, LISA struggles to identify *mashed potato*, possibly due to its transformed state from the raw ingredient. The second image presents a biscuit-based cake, where the model incorrectly focuses on crumbs rather than recognizing the entire structure as *biscuit*. The *Hanamaki Baozi* example represents an out-of-domain concept, similar to the previously discussed Worm-eating Warbler case, highlighting the model's limitations with unfamiliar items. In the salad image, LISA misinterprets individual vegetables as the salad itself, rather than recognizing the complete dish. Lastly, an adversarial example shows an apricot that visually resembles an egg, causing the model to fail in producing any output. This highlights LISA's vulnerability to visual similarities that deviate from expected appearances within a class. These examples illustrate the ongoing challenges in visual recognition tasks, particularly when dealing with transformed ingredients, culturally specific items, composite dishes, and visually ambiguous subjects.

Figure 4 presents additional visual examples of the top 10 classes that posed challenges for LISA. The *hair* class consistently proves problematic, with LISA often predicting the entire person instead of isolating the hair. For *upper clothes*, the model's misinterpretation can be attributed to linguistic ambiguity; in this instance, LISA incorrectly identified headwear as upper clothing, despite it being more accurately classified as an accessory. In the *soy* example, LISA fails to segment the soybean, instead erroneously detecting meatballs. The *tea* image shows the model including the cup in its segmentation rather than isolating the liquid alone. The final example demonstrates partial success, with LISA correctly identifying some cashews. However, it also exhibits a strong bias towards detecting non-relevant vegetables, leading to over-segmentation.

## C  EXTENDED QUANTITATIVE ANALYSIS

Tables 6 and 7 present comprehensive results for text prompted and vision-only models on MESS datasets, respectively. Table 8 shows oracle results, while Table 9 displays TP-VP framework outcomes.

| | Dataset | SEEM txt | CAT-Seg | Florence | PALI-Gem | LISA | Supervised |
|---|---|---|---|---|---|---|---|
| General | ATLANTIS | 48.4 | 30.5 | 14.4 | 46.8 | 63.9 | 45.1 |
| | BDD100K | 32.6 | 30.6 | 4.5 | 25.9 | 78.0 | 82.3 |
| | Dark Zurich | 33.1 | 45.8 | 11.4 | 21.8 | 41.1 | 44.8 |
| | DRAM | 60.4 | 33.6 | 29.3 | 58.6 | 78.6 | 42.2 |
| | FoodSeg103 | 31.0 | 30.0 | 18.1 | 51.3 | 60.6 | 53.2 |
| | MHP v1 | 10.0 | 33.1 | 6.5 | 7.6 | 19.8 | 63.9 |
| Earth | FloodNet | 59.6 | 9.2 | 28.6 | 62.5 | 72.9 | 84.6 |
| | iSAID | 9.5 | 66.5 | 4.1 | 4.3 | 31.3 | 45.7 |
| | ISPRS Potsdam | 40.7 | 53.9 | 11.0 | 23.9 | 41.0 | 74.0 |
| | UAVid | 57.5 | 39.0 | 11.5 | 34.7 | 59.8 | 87.2 |
| | WorldFloods | 16.9 | 16.1 | 14.4 | 20.3 | 33.4 | 65.3 |
| Medical | CHASE DB1 | 9.8 | 49.9 | 9.1 | 8.9 | 16.7 | 92.7 |
| | CryoNuSeg | 24.1 | 39.8 | 6.7 | 24.2 | 31.9 | 82.2 |
| | Kvasir-Inst. | 28.6 | 51.4 | 10.2 | 44.9 | 23.2 | 87.6 |
| | PAXRay-4 | 53.1 | 42.0 | 26.7 | 35.7 | 54.9 | 67.8 |
| Engin. | Corrosion CS | 11.1 | 25.0 | 7.7 | 8.8 | 13.8 | 97.1 |
| | DeepCrack | 4.2 | 35.1 | 5.5 | 4.5 | 6.8 | 73.5 |
| | PST900 | 14.3 | 79.4 | 6.3 | 2.9 | 12.1 | 93.7 |
| | ZeroWaste-f | 26.2 | 54.5 | 9.8 | 12.9 | 18.5 | 93.8 |
| Agri. | CUB-200 | 89.0 | 31.4 | 0.0 | 68.2 | 88.1 | 85.9 |
| | CWFID | 13.7 | 25.3 | 4.2 | 7.0 | 36.6 | 52.5 |
| | SUIM | 31.0 | 16.9 | 18.7 | 44.9 | 67.2 | 49.9 |

Table 6: Per dataset performance of text prompted methods

| | Dataset | SEEM vis | DINOv | VP | SoftMatcher+ | Supervised |
|---|---|---|---|---|---|---|
| General | ATLANTIS | 15.8 | 52.8 | 45.0 | 50.5 | 45.1 |
| | BDD100K | 7.2 | 37.8 | 53.1 | 57.8 | 82.3 |
| | Dark Zurich | 4.0 | 22.6 | 45.4 | 52.3 | 44.8 |
| | DRAM | 13.4 | 73.6 | 55.9 | 63.0 | 42.2 |
| | FoodSeg103 | 11.8 | 28.3 | 54.0 | 58.9 | 53.2 |
| | MHP v1 | 5.6 | 9.5 | 34.6 | 42.0 | 63.9 |
| Earth | FloodNet | 41.6 | 59.9 | 56.7 | 59.0 | 84.6 |
| | iSAID | 2.2 | 4.3 | 22.8 | 19.2 | 45.7 |
| | ISPRS Potsdam | 13.0 | 24.2 | 41.2 | 45.8 | 74.0 |
| | UAVid | 15.5 | 34.5 | 32.7 | 37.4 | 87.2 |
| | WorldFloods | 11.9 | 17.3 | 16.4 | 14.6 | 65.3 |
| Medical | CHASE DB1 | 10.4 | 9.6 | 0.0 | 0.0 | 92.7 |
| | CryoNuSeg | 26.8 | 24.0 | 21.2 | 21.6 | 82.2 |
| | Kvasir-Inst. | 6.5 | 24.4 | 65.7 | 59.9 | 87.6 |
| | PAXRay-4 | 38.1 | 39.0 | 39.0 | 52.2 | 67.8 |
| Engin. | Corrosion CS | 9.3 | 10.1 | 7.2 | 9.3 | 97.1 |
| | DeepCrack | 3.6 | 4.5 | 30.7 | 39.2 | 73.5 |
| | PST900 | 4.5 | 4.8 | 16.4 | 28.6 | 93.7 |
| | ZeroWaste-f | 10.4 | 13.9 | 21.0 | 25.2 | 93.8 |
| Agri. | CUB-200 | 20.7 | 92.2 | 85.4 | 87.0 | 85.9 |
| | CWFID | 17.5 | 33.5 | 41.5 | 33.3 | 52.5 |
| | SUIM | 26.9 | 51.4 | 52.5 | 58.9 | 49.9 |

Table 7: Per dataset performance of visual prompted methods

| | Dataset | SoftMatcher+ | LISA | Oracle | Oracle+ | Supervised |
|---|---|---|---|---|---|---|
| General | ATLANTIS | 51.4 | 63.9 | 63.9 | 68.9 | 45.1 |
| | BDD100K | 58.5 | 78.0 | 78.0 | 79.2 | 82.3 |
| | Dark Zurich | 47.7 | 41.1 | 47.7 | 55.0 | 44.8 |
| | DRAM | 62.9 | 78.6 | 78.6 | 81.3 | 42.2 |
| | FoodSeg103 | 60.5 | 60.6 | 60.6 | 74.0 | 53.2 |
| | MHP v1 | 36.7 | 19.8 | 36.7 | 45.3 | 63.9 |
| Earth | FloodNet | 57.4 | 72.9 | 72.9 | 74.8 | 84.6 |
| | iSAID | 26.7 | 31.3 | 31.3 | 35.4 | 45.7 |
| | ISPRS Potsdam | 41.4 | 41.0 | 41.4 | 50.2 | 74.0 |
| | UAVid | 35.7 | 59.8 | 59.8 | 65.0 | 87.2 |
| | WorldFloods | 20.0 | 33.4 | 33.4 | 33.4 | 65.3 |
| Medical | CHASE DB1 | 0.0 | 16.7 | 16.7 | 16.7 | 92.7 |
| | CryoNuSeg | 24.5 | 31.9 | 31.9 | 34.5 | 82.2 |
| | Kvasir-Inst. | 58.0 | 23.2 | 58.0 | 72.0 | 87.6 |
| | PAXRay-4 | 39.1 | 54.9 | 54.9 | 61.7 | 67.8 |
| Engin. | Corrosion CS | 14.8 | 13.8 | 14.8 | 17.6 | 97.1 |
| | DeepCrack | 39.3 | 6.8 | 39.3 | 42.2 | 73.5 |
| | PST900 | 38.9 | 12.1 | 38.7 | 39.7 | 93.7 |
| | ZeroWaste-f | 21.9 | 18.5 | 21.9 | 30.5 | 93.8 |
| Agri. | CUB-200 | 87.0 | 88.1 | 88.1 | 90.5 | 85.9 |
| | CWFID | 41.0 | 36.6 | 41.0 | 48.4 | 52.5 |
| | SUIM | 54.1 | 67.2 | 67.2 | 75.2 | 49.9 |

Table 8: Per dataset performance of Oracle ensembling baselines.

| | Dataset | SEEM | LISA | SoftMatcher+ | PromptMatcher | Oracle+ | Supervised |
|---|---|---|---|---|---|---|---|
| General | ATLANTIS | 15.8 | 63.9 | 51.4 | 55.7 | 68.9 | 45.1 |
| | BDD100K | 6.9 | 78.0 | 58.5 | 67.3 | 79.2 | 82.3 |
| | Dark Zurich | 4.3 | 41.1 | 47.7 | 51.7 | 55.0 | 44.8 |
| | DRAM | 13.5 | 78.6 | 62.9 | 69.7 | 81.3 | 42.2 |
| | FoodSeg103 | 12.0 | 60.6 | 60.7 | 61.9 | 74.0 | 53.2 |
| | MHP v1 | 5.8 | 19.8 | 36.7 | 46.2 | 45.3 | 63.9 |
| Earth | FloodNet | 40.7 | 72.9 | 57.4 | 61.4 | 74.8 | 84.6 |
| | iSAID | 2.3 | 31.3 | 26.7 | 24.3 | 35.4 | 45.7 |
| | ISPRS Potsdam | 13.1 | 41.0 | 41.4 | 45.9 | 50.2 | 74.0 |
| | UAVid | 14.9 | 59.8 | 35.7 | 52.4 | 65.0 | 87.2 |
| | WorldFloods | 14.2 | 33.4 | 20.0 | 14.7 | 33.4 | 65.3 |
| Medical | CHASE DB1 | 10.4 | 16.7 | 0.0 | 0.0 | 16.7 | 92.7 |
| | CryoNuSeg | 27.1 | 31.9 | 24.5 | 24.1 | 34.5 | 82.2 |
| | Kvasir-Inst. | 6.4 | 23.2 | 58.0 | 60.8 | 72.0 | 87.6 |
| | PAXRay-4 | 38.1 | 54.9 | 39.1 | 55.5 | 61.7 | 67.8 |
| Engin. | Corrosion CS | 10.4 | 13.8 | 14.8 | 15.2 | 17.6 | 97.1 |
| | DeepCrack | 3.8 | 6.8 | 39.3 | 42.6 | 42.2 | 73.5 |
| | PST900 | 4.9 | 12.1 | 38.9 | 39.3 | 39.9 | 93.7 |
| | ZeroWaste-f | 10.1 | 18.5 | 21.9 | 24.6 | 30.5 | 93.8 |
| Agri. | CUB-200 | 21.1 | 88.1 | 87.0 | 88.9 | 90.5 | 85.9 |
| | CWFID | 17.5 | 36.6 | 41.0 | 38.4 | 48.4 | 52.5 |
| | SUIM | 28.8 | 67.2 | 54.1 | 59.8 | 75.2 | 49.9 |

Table 9: Per dataset performance of visual-text prompted methods

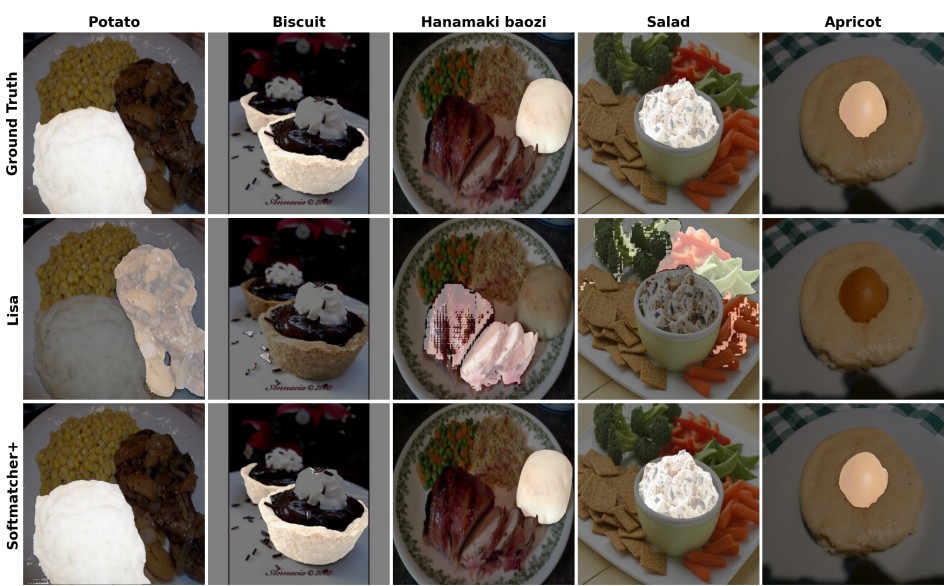

Figure 3: Qualitative examples selected from the most challenging classes of FoodSeg103.

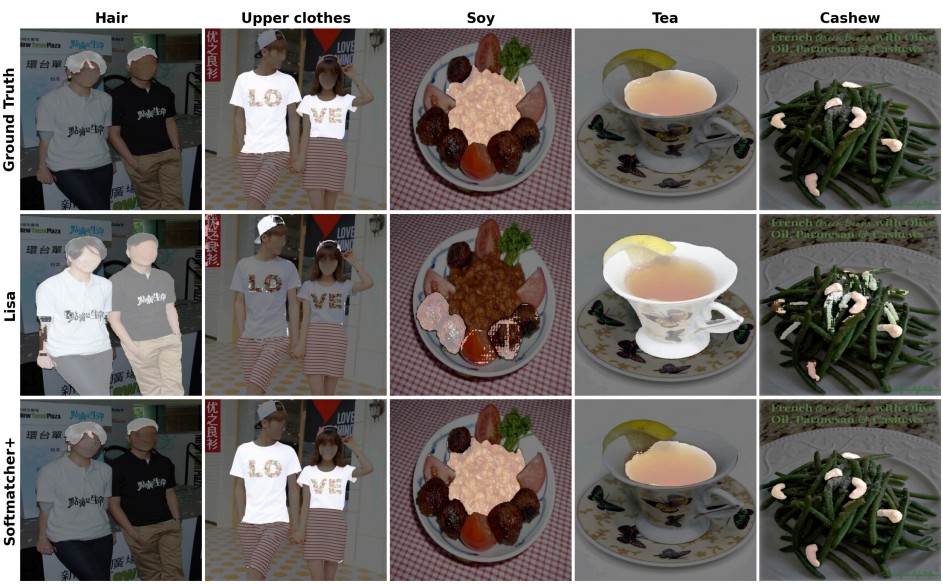

Figure 4: Qualitative analysis on examples of challenging classes for Text Prompting.