# OpenReview forum: "Show or Tell? Effectively prompting Vision-Language Models for semantic segmentation"
_ICLR.cc/2025/Conference — ICLR 2025 Conference Withdrawn Submission_

### Official Review · Reviewer_WrP1 · 2024-10-26

**Soundness:** 2
**Presentation:** 3
**Contribution:** 2
**Rating:** 5
**Confidence:** 4

**Summary:**

This paper investigates how to effectively prompt Large Vision-Language Models (VLMs) for semantic segmentation using the MESS dataset.  The authors introduce few-shot prompted semantic segmentation and discover that text and visual prompts are complementary. The paper also proposes PromptMatcher, a simple baseline combining both prompt types, achieving considerable results in training-free semantic segmentation.

**Strengths:**

1. The paper introduces a novel benchmarking task to evaluate the performance of Large Vision-Language Models (VLMs) for semantic segmentation, providing a systematic assessment of their capabilities and limitations.

2. The  paper shows that text and visual prompting complement each other, and propose a simple training-free framework to capitalize on the complementary strengths of text and visual prompts.

**Weaknesses:**

1. The proposed PromptMatcher appears to be a simple combination of SoftMatch [2] and LISA [1]. The complementary nature of visual prompts and text prompts has already been extensively described in these previous works, which means that the innovation in this paper is somewhat limited.

2. Recent work on Large Language Models (LLMs) for segmentation [3] has already demonstrated superior performance in certain aspects compared to previous expert models. However, the paper does not provide sufficient discussion on these advancements, making the claim that "the latest models remain far below custom models trained for a specific task and data domain" somewhat debatable.

3. There are issues with the experimental comparisons presented in the paper. For instance, in Table 4, LISA shows significantly better performance on the Earth task compared to PromptMatcher. Additionally, the overall improvement in average performance by PromptMatcher is relatively minor.

[1] LISA: Reasoning Segmentation via Large Language Model

[2] SoftMatch: Addressing the Quantity-Quality Trade-off in Semi-supervised Learning

[3] LLaFS: When Large Language Models Meet Few-Shot Segmentation

**Questions:**

Please reder to the Weakness

---

### Official Review · Reviewer_Zrz2 · 2024-10-27

**Soundness:** 2
**Presentation:** 2
**Contribution:** 1
**Rating:** 3
**Confidence:** 4

**Summary:**

This paper introduces a new task setting: few-shot prompted semantic segmentation, to explore the performance of VLMs guided by text and visual prompts. The authors found that text and visual prompts are complementary. Based on this finding, they combined text and visual prompts, proposed PromptMatcher, and achieved state-of-the-art performance.

**Strengths:**

S1: This paper is well written and easy to follow.

S2: This paper proposes a new task, few-shot prompted semantic segmentation, to explore VLMs and conducts a thorough experimental analysis.

S3: The paper finds that text and visual prompts are complementary, and based on this, proposes a combined approach that achieves significant performance improvement.

**Weaknesses:**

W1: Limited originality. Although the complementary nature of text and visual prompts discussed in this paper is interesting, the proposed PromptMatcher appears to be merely a simple combination of existing methods (LISA and SoftMatcher+). This is an engineering combination with a lack of methodological contribution.

W2: Insufficient experiments. The paper is only validated on the MESS dataset, while experiments on more general datasets (such as COCO, PASCAL, RefCOCO, etc.) are necessary. These experiments could provide stronger support for the paper's conclusions.

Overall, this work appears to be in an early stage in terms of both methodology and experimentation. The authors are encouraged to explore commonalities across different models more thoroughly and to combine prompts from various modalities in a way that distills the essential structure rather than simply combining existing models. To guide this exploration, I suggest conducting an ablation study to determine which components of LISA and SoftMatcher+ contribute most to the observed performance gains. Additionally, the authors could explore techniques like knowledge distillation to create a more unified model, enhancing the theoretical contribution and overall robustness of their approach.

**Questions:**

see weaknesses

---

### Official Review · Reviewer_XkdU · 2024-11-03

**Soundness:** 3
**Presentation:** 3
**Contribution:** 3
**Rating:** 3
**Confidence:** 4

**Summary:**

This paper addresses the main prompt-based semantic segmentation methods, one is based on text prompts and another one is based on visual prompts, each of which exhibits distinct advantages in different scenarios. The authors observe that these two approaches have complementary effects in certain scenarios. Therefore the authors propose a combined prompt scheme that integrates both approaches, achieving good results on benchmarks of training-free semantic segmentation.

**Strengths:**

- The author's observation is of practical significance, as text prompt-based and visual prompt-based segmentation methods should indeed be combined in different scenarios to achieve better results.
- This paper is well-written and easy to follow.
- The author has validated the effectiveness of the proposed method.

**Weaknesses:**

- **Limited Novelty**: The author's observation has already been discussed in previous papers, like T-Rex2 [1]. Much of the article simply restates this conclusion for semantic segmentation without offering more insightful observations. Moreover, the method proposed by the author is a simple combination of the existing works SoftMatcher+ and LISA, which has limited innovation.
- **Limited Scope and Generalizability of the Model**: The paper primarily conducts experiments within the scope of semantic segmentation. However, in real-world applications, tasks like panoptic segmentation and instance segmentation may be more relevant. The proposed approach has limited generalizability and may not extend effectively to other segmentation scenarios, such as panoptic and instance segmentation.
- **Limited Performance Improvement**: The proposed method aims to reduce the performance gap between the Oracle Ensemble & Oracle Ensemble+ methods (combining the best results from text and visual prompts) and using only visual or text prompts individually. However, the PromptMatcher still shows a significant gap from Oracle Ensemble+ (45.3 vs. 53.8). Given that varying the text prompt can substantially impact model performance, it raises the question of whether more precise refinement of the text prompt could further enhance the performance of this pipeline.

References:

[1] [T-Rex2: Towards Generic Object Detection via Text-Visual Prompt Synergy (ECCV 2024)](https://arxiv.org/abs/2403.14610)

**Questions:**

Please see the weaknesses above.

---

### Official Review · Reviewer_Ry3E · 2024-11-04

**Soundness:** 2
**Presentation:** 2
**Contribution:** 2
**Rating:** 3
**Confidence:** 4

**Summary:**

This paper explores how to effectively prompt large VLMs for semantic segmentation by systematically evaluating text and visual prompts on the MESS dataset. It benchmarks some visual prompt and text prompt models on the dataset. It proposes a simple boosting technique by combining two different methods.

**Strengths:**

- The paper is straightforward and easy to understand. The proposed method is reasonable and logically structured.
- The paper includes a detailed analysis of model behavior in some sections.

**Weaknesses:**

High-level weaknesses:
- The first claimed contribution states, “We design a novel benchmarking task to probe the performance of VLMs as semantic segmentation engines.” However, this benchmark appears identical to the one introduced in the MESS paper, which also evaluates more models in both its paper and associated webpage. Therefore, it may be inaccurate to present this as a novel benchmarking effort. Additionally, it would strengthen the paper to include models proposed in the MESS paper for comparison.
- The paper's main contributions seem to focus on (1) benchmarking models on the MESS dataset and (2) proposing a simple ensembling method. However, there is a lack of direct comparison to established benchmarks like MESS and an absence of ablation studies to validate the proposed method. The conclusions drawn from the results are shallow and straightforward, limiting the insights provided.
- For example, for the proposed PromptMatcher, the readers are interested in the contribution of each components, like the point proposals, rejection models, or influence of different text prompts.These ablations are totally missed in the paper.
- The third claimed contribution, stating “We show that text and visual prompting complement each other, and that being able to anticipate the most effective prompt modality can lead to an 11% improvement in performance,” does not align with the results. As Table 4 shows, the PromptMatcher only improves by 2.5% over the previous SOTA, LISA. Citing oracle results as representative of the paper's impact is misleading.
- The title of the paper suggests it explores prompting for VLMs; however, the proposed method primarily connects VLMs with other models, as evidenced by the PromptMatcher’s design. The title may therefore misrepresent the actual scope of the work.

Detailed weaknesses:
- The caption for Table 1 is inconsistent with the table’s structure. The first block should indicate text-prompt methods, while the second block should cover visual prompts. This order is reversed in the caption, leading to confusion.
- There are discrepancies in results across tables: for example, SoftMatcher+ achieves a score of 41.6 in Table 1 but 41.8 in Table 2. This inconsistency could raise questions about the article’s credibility.
- Similarly, the results of CAT-Seg are different from the results in MESS, which needs more clarification.

**Questions:**

NA

---

### Author Response · Authors · 2024-11-21

After careful reflection, we have decided to withdraw our submission to focus on further developing and refining our work. We sincerely appreciate the thoughtful feedback and insights provided by the Reviewers and recognize the importance of delivering our message more clearly and effectively, and we value the constructive comments that will guide this process. Thank you once again for your time and thorough evaluation.

---

### Note · Authors · 2024-11-21

I have read and agree with the venue's withdrawal policy on behalf of myself and my co-authors.